# Exercise Intensity during Olympic-Distance Triathlon in Well-Trained Age-Group Athletes: An Observational Study

**DOI:** 10.3390/sports9020018

**Published:** 2021-01-21

**Authors:** Atsushi Aoyagi, Keisuke Ishikura, Yoshiharu Nabekura

**Affiliations:** 1Graduate School of Comprehensive Human Sciences, University of Tsukuba, 1-1-1 Tennodai, Tsukuba, Ibaraki 305-8574, Japan; atsushi.aoyagi.1992@gmail.com; 2Faculty of Management, Josai University, 1-1 Keyakidai, Sakado, Saitama 350-0295, Japan; ishikura@josai.ac.jp; 3Faculty of Health and Sport Sciences, University of Tsukuba, 1-1-1 Tennodai, Tsukuba, Ibaraki 305-8574, Japan

**Keywords:** multisport, endurance performance, intensity profile, swimming, cycling, running, heart rate, aerobic threshold, anaerobic threshold, workload

## Abstract

The aim of this study was to examine the exercise intensity during the swimming, cycling, and running legs of nondraft legal, Olympic-distance triathlons in well-trained, age-group triathletes. Seventeen male triathletes completed incremental swimming, cycling, and running tests to exhaustion. Heart rate (HR) and workload corresponding to aerobic and anaerobic thresholds, maximal workloads, and maximal HR (HR_max_) in each exercise mode were analyzed. HR and workload were monitored throughout the race. The intensity distributions in three HR zones for each discipline and five workload zones in cycling and running were quantified. The subjects were then assigned to a fast or slow group based on the total race time (range, 2 h 07 min–2 h 41 min). The mean percentages of HR_max_ in the swimming, cycling, and running legs were 89.8% ± 3.7%, 91.1% ± 4.4%, and 90.7% ± 5.1%, respectively, for all participants. The mean percentage of HR_max_ and intensity distributions during the swimming and cycling legs were similar between groups. In the running leg, the faster group spent relatively more time above HR at anaerobic threshold (AnT) and between workload at AnT and maximal workload. In conclusion, well-trained male triathletes performed at very high intensity throughout a nondraft legal, Olympic-distance triathlon race, and sustaining higher intensity during running might play a role in the success of these athletes.

## 1. Introduction

Triathlon is a multidisciplinary endurance sport consisting of swimming, cycling, and running over a variety of distances [1]. The most common distances include the sprint (25.75 km, ~1 h), half-Ironman (113 km, ~4–5 h), and Ironman (226 km, ~8–17 h), and the most popular is the so-called Olympic distance (OD), consisting of standard distances for swimming (1.5 km), cycling (40 km), and running (10 km), for a total of 51.5 km. Scientific interest in the triathlon has significantly increased since the introduction of the 51.5-km race in the 2000 Summer Olympics, held in Sydney [2]. The total competition time of the OD race ranged from about 1 h 50 min to 2 h 40 min, with swimming accounting for 16%–19% (20–30 min), cycling of about 50%–55% (60–80 min), and running around 29%–31% (30–50 min) [3,4].

It is necessary to discern the physiological response and requirements during competition to optimize training and recovery, and to identify factors associated with performance. The efficient transition between two sequential legs of the triathlon has received considerable attention, e.g., the swim–cycle [5,6] and cycle–run [7,8] transitions have been well studied. Additionally, previous studies have assessed the acute consequences induced by the OD race, which include muscle [9,10] and intestinal damage [9], muscle fatigue [10], dehydration (>2%–4% body mass due to high sweat rates and high core temperatures, i.e., >39 °C) [11,12], systemic inflammation [13], transient immune suppression [14], reduced pulmonary diffusing capacity [15], and decreased aerobic exercise capacity [16,17].

Although a large number of studies have investigated the physiological responses and acute consequences of multidisciplinary and endurance sporting events, relatively few have addressed the sustained exercise intensity encountered during an actual OD race. Oxygen consumption ($\dot{V}$O_2_) and blood lactate concentration (BLa) are two of the main parameters used to quantify exercise intensity [18]. However, it is difficult to measure these variables during actual competition. Over the past two decades, heart rate (HR), as a marker of internal load, has been used to estimate exercise intensity [19,20], by relating individual competition HR values measured in the field with those obtained in a laboratory incremental test [20,21,22]. In addition, the combination of internal load (HR) and external load (workload, i.e., speed and power output (PO)) can provide important information about the physiological demands during endurance events [19]. Thus, knowledge of exercise intensity profiles based on internal and external loads can facilitate greater comprehension of the physiological demands of the OD triathlon.

Studies investigating exercise intensity during an actual OD race are sparse. According to Bernard et al. [21], the mean relative HR and workload of elite triathletes during cycling in OD race were 91% ± 4% of the maximal heart rate (HR_max_) and 60% ± 8% of the maximal aerobic power (MAP), although these measurements were not reported during the swimming and running legs. A study conducted by Le Meur et al. [22] of the mean relative HR and workload during an OD race in relation to the individual metabolic capacities of elite triathletes assessed at each exercise mode (i.e., swimming, cycling, and running) reported 91%–92% of HR_max_ for swimming, 90%–91% of HR_max_ and 61.4%–63.4% of MAP for cycling, and 93%–94% of HR_max_ for running. However, the cohorts of these studies [21,22] were limited to elite triathletes. While elite triathletes compete in draft legal cycling, in which a competitor is permitted to draft within a sheltered position behind another, nonelite or age-group triathletes usually compete in nondraft legal racing. Drafting directly affects exercise intensity by reducing $\dot{V}$O_2_ (−14%) and HR (−7%), as compared to nondraft cycling with the same external load (speed) by triathletes [23]. Hence, it is necessary to distinguish between draft legal and nondraft legal races. To the best of our knowledge, only two studies have reported the exercise intensity sustained during actual nondraft legal OD races [24,25]. However, these were limited by reporting the absolute values of both HR (bpm) and workload (swimming speed, km·h^−1^; cycling PO, W; and running speed, km·h^−1^) [24] or the relative HR, but not workload, during cycling and running (not swimming), similar to an OD race (swimming:, 1.0 km; cycling, 30 km; and running, 8 km) [25]. Therefore, no study to date has investigated the relative exercise intensity sustained and the distribution of intensity during the entire duration of an actual nondraft legal OD race.

Furthermore, in other endurance events, there is an obvious tendency toward reduced relative intensity in relation to increased race duration. In ultra-endurance events of more than 8 h, such as the Ironman triathlon [26] and the 65-km run of a mountain ultra-marathon [27], the relative intensity during competition was lower than the ventilatory threshold (VT) and 80% of the HR_max_. Hence, these intensities have been proposed as an “ultra-endurance threshold” [6]. In the 42-km marathon (~2.5–5 h), the mean relative HR is reportedly around 80%–90% of HR_max_ [28,29], which is similar to the HR at VT [30]. During a shorter event, such as a 5–10 km running (~15–55 min), the mean HR values are reportedly higher (~90–96% of HR_max_) [28]. On the other hand, the relative intensity that can be sustained during an endurance race may also be related to differences in performance levels [31,32]. Thus, even among triathletes, the relative intensity during an OD triathlon could also differ, depending on the exercise duration and/or performance level.

Therefore, the aim of this study was (i) to estimate, using competition HR and workload data, the relative exercise intensity in all three disciplines during a nondraft legal OD race in well-trained, age-group triathletes, and (ii) to compare the estimated intensity of fast and slow triathletes. We hypothesized that triathletes would maintain a high level of intensity throughout the race, and that faster triathletes could perform the race at higher relative intensity than slower triathletes.

## 2. Materials and Methods

### 2.1. Study Design

This study was conducted in two phases consisting of laboratory tests and during-race monitoring. The laboratory tests included an incremental swimming test, an incremental cycling test, and an incremental treadmill running test, which were performed randomly and separated by a minimum of 2 days and maximum of 20 days. Competition measurements of each participant were conducted during the OD race with a nondraft legal cycling leg and were timed as close as possible to all laboratory tests (mean ± standard deviation, 45 ± 26 days).

### 2.2. Subjects

The study cohort consisted of 17 well-trained, age-group male triathletes (Table 1) who met the following inclusion criteria: (1) regular training of at least five sessions per week for a triathlon competition; (2) not suffering from any present injury, which could have possibly hampered their performance, and nonsmokers; and (3) a minimum of one year of experience competing in triathlons. The median time for completion of the OD race based on pooled data was 2:16:13 h:min:s. The subjects were split into two groups according to the total time of the OD race. Participants with times not less than 2:16:13 were assigned to the faster group and those with times greater than 2:16:13 was assigned to the slower group. The study protocol was approved by the Ethics Committee of the University of Tsukuba (project identification code: Tai 30–24) and conducted in accordance with the ethical principles for medical research involving human subjects as described in the Declaration of Helsinki. All subjects provided written informed consents to participate in the study.

### 2.3. Laboratory Tests

The subjects were instructed to refrain from consuming caffeine and alcohol and from heavy training on the day before the tests, as well as to consume a light meal at least 3 h before each laboratory test. During the 3-h period preceding the tests, only ad libitum water ingestion was permitted. During all laboratory tests, HR was collected via a HR monitor (HRM-Tri; Garmin Ltd., Olathe, KS, USA) across the chest with sampling at 1 Hz. Body mass was measured with a body fat monitor scale (TBF-102; Tanita, Tokyo, Japan) before the cycling and running tests.

#### 2.3.1. Incremental Swimming Test

The subjects completed a two-part test consisting of a submaximal intermittent test and a maximal incremental test that were performed in a swimming flume at a constant water temperature of 25.8 ± 0.8 °C. The subjects wore their own technical trisuits, standard swimming caps, and goggles throughout the incremental swimming test. The submaximal intermittent test was performed first. The swimming speed of the submaximal intermittent test was individualized according to the average swimming speed of the most recent 1500-m time trial (S_1500_) of each triathlete. The speed for the initial stage was 70% of S_1500_ and increased by 5% at each subsequent stage for a total five to seven stages, each consisting of 4 min of exercise and 2 min of rest. Before the test and after each stage, blood samples were obtained from the fingertip and the BLa was measured with a lactate analyzer (Lactate Pro 2; Arkray, Inc., Kyoto, Japan). The submaximal intermittent test was concluded when the BLa exceeded 4.0 mmol·L^−1^ or the rate of perceived exertion was ≥15. Following a 5-min recovery period after the submaximal intermittent test, the maximal incremental test was performed. The initial speed was set at that of the next to last stage of the submaximal intermittent test and then was increased by 0.03 m·s^−1^ every minute until volitional exhaustion, which was defined as the point at which the subject could no longer swim at the required speed.

#### 2.3.2. Incremental Cycling Test

The maximal incremental test was performed on an electronically braked, indoor cycle trainer (CompuTrainer Pro; RacerMate Inc., Seattle, WA, USA), which allowed the subjects to use their own bicycles, at a constant room temperature of 25.2 ± 1.2 °C, relative humidity of 40.4% ± 8.3%, and barometric pressure of 755.4 ± 3.5 mmHg with an electric fan ensuring air circulation around the participant. The maximal incremental test was performed following a 5-min warm-up period (100 W) and a 5-min recovery period. The initial workload was set at 100 W and then increased by 20 W·min^−1^ and cadence was maintained at 80 or 90 rpm in accordance with race cadence of the individual until volitional exhaustion, which was defined as <75 or 85 rpm (i.e., a decrease of 5 rpm as compared with the cadence of the sustained during test) continuously for 5 s. To determine the PO during the test, which was used for analysis, the bicycles were fitted with calibrated power measuring pedals (Garmin Ltd.) at a sampling rate of 1 Hz.

#### 2.3.3. Incremental Treadmill Running Test

The maximal incremental test was performed on a motorized treadmill (ORK-7000; Ohtake-Root Kogyo Co., Ltd., Iwate, Japan) at a grade of 1% to accurately reflect the energetic cost of outdoor running [33] following a 5-min warm-up period (9.0 km·h^−1^) and a 5-min recovery period. The experimental environmental conditions were similar to those of the incremental cycling test. The initial speed was set at 9.0 km·h^−1^ and then increased by 0.6 km·h^−1^ every minute until volitional exhaustion, which was defined as the inability of the subject to continue running at the required speed. The treadmill belt speed, which was used for analysis, was measured with a hand-held tachometer (EE-1B; Nidec-Shimpo Corporation, Kyoto, Japan).

### 2.4. Gas Analysis

During the incremental cycling and treadmill running tests, $\dot{V}$O_2_, carbon dioxide production ($\dot{V}$CO_2_), minute ventilation ($\dot{V}$_E_), ventilatory equivalent of oxygen ($\dot{V}$_E_/$\dot{V}$O_2_) and carbon dioxide ($\dot{V}$_E_/$\dot{V}$CO_2_), end-tidal partial pressure of oxygen (P_ET_O_2_) and carbon dioxide (P_ET_CO_2_), and respiratory exchange ratio (RER) were measured on a breath-by-breath basis using a computerized standard open circuit technique with a metabolic gas analyzer (AE-310s; Minato Medical Science Co., Ltd., Osaka, Japan).

Before both tests, the metabolic system was calibrated using known gas concentrations and a 2-L syringe in accordance with the manufacturer’s instructions. Maximal $\dot{V}$O_2_ ($\dot{V}$O_2 max_), which was defined as the highest 1-min rolling average (20-s × 3), was attained when at least two of the following four criteria were met: (1) a leveling-off of $\dot{V}$O_2_ despite an increase in PO or running speed, (2) peak RER ≥ 1.10, (3) peak HR ≥ 90% of age-predicted values, and (4) perceived exertion score at the end of the tests of ≥19. The gross efficiency during cycling was calculated as described in a previous study [34]. The running economy was expressed as the O_2_ cost (ml·kg^−1^·km^−1^) and was calculated based on the last 1-min $\dot{V}$O_2_ while running for 5 min (9.0 km·h^−1^) [35].

### 2.5. Determination of Aerobic and Anaerobic Thresholds, and Maximal Workload

The predicted swimming speed and HR at both the lactate threshold (LT) and the onset of blood lactate accumulation (OBLA) of 4 mmol·L^−1^ were calculated using validated software (Lactate-E [36], version 2.0, National University of Galway, Galway, Ireland) based on the BLa, swimming speed, and HR collected during the incremental swimming test.

VT during cycling and running was determined using the criteria of an increase in both $\dot{V}$_E_/$\dot{V}$O_2_ and P_ET_O_2_ with no increase in $\dot{V}$_E_/$\dot{V}$CO_2_, whereas the respiratory compensation point (RCP) was determined using the criteria of an increase in both $\dot{V}$_E_/$\dot{V}$O_2_ and $\dot{V}$_E_/$\dot{V}$CO_2_, and a decrease in P_ET_CO_2_ [37,38]. Two independent observers determined the VT and RCP for cycling and running. Any disagreement was mediated by the opinion of a third investigator [39].

Based on previous studies [40,41], the LT obtained from the swimming test and the VT obtained from the cycling and the treadmill running tests were defined as the “aerobic threshold” (AeT), while the OBLA of 4 mmol·L^−1^ obtained from the swimming test and the RCP obtained from cycling and running test were defined as the “anaerobic threshold” (AnT).

Maximal swimming speed (SS_max_) and maximal running speed (RS_max_) were determined by the last stage of the maximal incremental test. When the subject was unable to complete 1 min at the current workload, the maximal workload was determined by adding a fraction of the final workload to the workload of the immediately preceding 1 min [42]. Maximal PO (PO_max_) was determined as the highest 20-s rolling average attained during the incremental cycling test. These three variables were collectively referred to as the “maximal workload”.

### 2.6. Competition Measurements

The OD races were nonunified between each individual because of the difficulty of securing a sufficient sample size among the participants of the same race. The distance of each OD race was the same and consisted of swimming for 1.5 km, cycling for 40 km, and running for 10 km. During all races, the subjects wore a HR monitor (HRM-Tri) across the chest with sampling at 1 Hz. Also, the subjects wore waterproofed portable global navigation satellite system (GNSS) units (ForeAthlete 920XT; Garmin Ltd., Olathe, KS, USA) on the wrist with sampling at 1 Hz to determine the speed throughout the race. To determine cycling PO during the race, the bicycles were fitted with calibrated power measuring pedals (Garmin Ltd.) set at a sampling rate of 1 Hz. Following completion of each race, performance times were obtained from official websites of the event. Air temperature, relative humidity, wind speed, and barometric pressure were obtained from the local meteorological agency within 1 h of the start of the race. Water temperature was measured with a water temperature gauge (CTH-1365; Custom Corporation, Tokyo, Japan) before the start of the race. Cumulated positive elevation during cycling and running was obtained from the GNSS units. The elevation to distance ratio was calculated by dividing the cumulated positive elevation by 40 (km, for cycling) or 10 (km, for running).

### 2.7. Exercise Intensity Zone Settings

Based on the HR measurements, the total time of each leg of the OD race was divided into three intensity zones based on the results of the laboratory tests in the corresponding exercise mode. The percentage of time spent in each zone was calculated as follows: less than AeT (HR_zone_1), between AeT and AnT (HR_zone_2), more than AnT (HR_zone_3) [43,44]. Similar analysis of the PO measurements in the cycling leg [22,45] was also conducted: below 10% of PO_max_ (PO_zone_1), between 10% of PO_max_ and AeT (PO_zone_2), between AeT and AnT (PO_zone_3), between AnT and PO_max_ (PO_zone_4), and above PO_max_ (PO_zone_5). In addition, using the average running speed (RS) at each 100 m in the running leg, an analysis similar to that for PO distribution was conducted based on the incremental running test results, rather than the incremental cycling test results.

### 2.8. Data Analysis

HR, PO, and speed obtained from the GNSS units were recorded with the GNSS units (Garmin Ltd.). The obtained data were exported to third party, open-source analysis software (Golden Cheetah, version 3.4, http://www.goldencheetah.org/), and further analysis was performed using Microsoft Excel software (version 2019, Microsoft Corporation, Redmond, WA, USA). Expired gas data were averaged across 20-s intervals using internal gas analysis software (version 3STG, AT Windows; Minato Medical Science Co., Ltd., Osaka, Japan), and exported to a.csv file. Further analysis was performed using Microsoft Excel 2019 software.

### 2.9. Statistical Analysis

All results are presented as the mean ± standard deviation unless otherwise indicated. Normal distribution of the data was tested using the Kolmogorov–Smirnov test. The nonparametric Mann–Whitney U test was used to detect statistically significant differences between groups. The effect size (ES) was evaluated as *r* (with 0.1 considered to be a small, 0.3 a medium, and 0.5 a large effect [46]). All statistical analyses were conducted using IBM SPSS Statistics version 26 (IBM Japan, Tokyo, Japan). A probability (*p*) value of ≤0.05 was considered statistically significant.

## 3. Results

The competition measurement data of the OD race were corrected from 10 races conducted in 2018 or 2019. The environmental conditions during the OD races are shown in Table 2. There were no significant differences observed between the two groups (all, *p* ≥ 0.16; ES = 0.04–0.35). During the swimming leg of the race, most of the subjects wore their own wetsuits, with the exception one subject in each group who exercised in nonwetsuits.

### 3.1. Laboratory Tests

The results of the three incremental tests of each exercise mode are summarized in Table 3. The faster group (*n* = 9) was superior to the slower group (*n* = 8) in speed at AeT and AnT and SS_max_ for the swimming test (*p* < 0.01, respectively, ES = 0.68–0.76), and PO at AeT (*p* = 0.04, ES = 0.51), PO_max_ (*p* = 0.03, ES = 0.53), and $\dot{V}$O_2 max_ (L·min^−1^ and ml·kg^−1^·min^−1^, *p* = 0.01 and < 0.01, respectively, ES = 0.61, 0.72, respectively) for the cycling test. However, there were no significant differences in the treadmill running test results between groups (all, *p* ≥ 0.14; ES = 0.00–0.36).

### 3.2. OD Triathlon Performance

The OD race performance times are shown in Table 1. The mean total performance time of the OD race was 2:19:50 ± 0:09:38 (h:min:s). The faster group had a shorter time than the slower group (2:12:24 ± 0:02:54 vs. 2:28:12 ± 0:07:11, *p* < 0.01, ES = 0.84). The time of all three legs was shorter for the faster group than the slower group (all, *p* ≤ 0.02; ES = 0.58–0.82).

### 3.3. Exercise Intensity of Each Leg

The mean exercise intensity of each leg of the OD race is shown in Table 4. The overall mean absolute workload was 1.03 ± 0.18 m·s^−1^ for swimming, 209.6 ± 24.3 W for cycling, and 14.1 ± 1.4 km·h^−1^ for running. The absolute workload was greater in the faster group than the slower group for swimming (1.14 ± 0.18 vs. 0.92 ± 0.09 m·s^−1^, respectively, *p* < 0.01, ES = 0.68) and running (14.9 ± 0.8 vs. 13.2 ± 1.4 km·h^−1^, respectively, *p* = 0.02, ES = 0.58). The absolute workload for cycling was also greater in the faster group than the slower group, although the difference was not significant (*p* = 0.06, ES = 0.47). The mean relative HR of all participants was 89.8% ± 3.7% of HR_max_ for swimming, 91.1% ± 4.4% of HR_max_ for cycling, and 90.7% ± 5.1% of HR_max_ for running. There were no differences in mean relative HR between groups for swimming and cycling (*p* = 0.25 and 0.07, respectively, ES = 0.28, 0.44, respectively). However, the relative HR for running of the faster group was greater than that of the slower group (*p* < 0.01, ES = 0.65). The relative workloads and absolute HR values of each leg are shown in Table 4.

The absolute workloads and relative HR values (i.e., % of HR_max_) of each leg during the OD race of all participants are shown in Figure 1. The average AeT, AnT, and maximal values of each laboratory test of each leg are shown for visual reference. The mean workload value shifted above workload at AnT in swimming, between workload at AeT and AnT in cycling, and slightly below workload at AnT, with the exception of the beginning and end of the leg, in running. The mean relative HR value remained above HR at AnT throughout the race.

### 3.4. Exercise Intensity Distribution during the Race

During the swimming, cycling, and running legs of the OD race, the HR distribution of all participants was 1.5% ± 2.3%, 2.8% ± 8.0%, and 4.1% ± 10.6% in HR_zone_1, 6.6% ± 15.0%, 18.4% ± 24.0%, and 39.9% ± 38.5% in HR_zone_2, and 91.9% ± 16.3%, 78.8% ± 28.1%, and 56.0% ± 42.1% in HR_zone_3, respectively. There were no significant differences in HR distribution between groups in the swimming and cycling legs (all, *p* ≥ 0.20; ES = 0.14–0.32; Figure 2). In running, however, the slower group had a higher percentage in HR_zone_2 (65.4% ± 35.0% vs. 17.3% ± 25.9%, *p* = 0.02, ES = 0.58) and a lower percentage in HR_zone_3 (26.2% ± 36.4% vs. 82.4% ± 26.6%, *p* = 0.01, ES = 0.60) as compared to the faster group (Figure 2).

The PO distribution of all participants was 5.8% ± 3.2% in PO_zone_1, 30.0% ± 18.1% in PO_zone_2, 36.3% ± 17.2% in PO_zone_3, 22.8% ± 11.1% in PO_zone_4, and 5.1% ± 2.8% in PO_zone_5, respectively. There were no significant differences in PO distribution between the two groups (all, *p* ≥ 0.14; ES = 0.09–0.35; Figure 3A). The RS distribution of all participants was 0.0% ± 0.0% in RS_zone_1, 8.7% ± 12.5% in RS_zone_2, 52.1% ± 28.9% in RS_zone_3, 38.8% ± 31.0% in RS_zone_4, and 0.5% ± 0.7% in RS_zone_5, respectively. As compared to the slower group, the faster group had a higher percentage only in RS_zone_4 (53.8% ± 31.7% vs. 21.9% ± 20.6%, *p* = 0.04, ES = 0.50) (Figure 3B).

## 4. Discussion

To the best of our knowledge, this study is the first to evaluate the relative exercise intensity of well-trained, age-group male triathletes at two performance levels according to the principle of test specificity in all disciplines during a nondraft legal OD race. The main findings of this study were as follows: (1) Well-trained, age-group male triathletes who completed the nondraft legal OD at a mean time (h:min:s) of 2:19:50 ± 0:09:38 (range, 2:07:16–2:41:07) demonstrated a high percentage of HR_max_ (87% of HR_max_~) throughout all three legs; (2) Faster triathletes had shorter times in all three legs with higher absolute workloads, but differences in relative intensity and intensity distribution were observed only in the running leg, as faster triathletes sustained higher intensity than slower triathletes.

### 4.1. Laboratory Tests and OD Triathlon Performance

The total performance times of the OD race in this study were close to those obtained in other studies with similar well-trained triathletes [47], but shorter than a study of recreational triathletes [48]. It is well known that successful endurance athletes are characterized by high levels of maximal ($\dot{V}$O_2 max_ and maximal workload) and submaximal (aerobic/anaerobic threshold and economy/efficiency) measures [49,50]. In the present study, the times to complete the swimming (−20.8%) and cycling (−4.9%) legs in an actual OD race were significantly shorter for the faster group in association with superior maximal and submaximal measures in the incremental swimming and cycling tests, as compared with the slower group. However, the time to complete the running leg was significantly shorter for the faster group (−13.2%), while there was no significant difference in any parameters in the incremental running test between the groups. These results suggest that the differences in swimming and cycling performance between the two groups were highly dependent on aerobic capacities, as assessed by specific incremental tests for each of the exercise modes. However, the difference in running performance between the two groups may have been due to factors other than aerobic capacities, as assessed by the incremental treadmill running test in this study. A review article [7] pointed out that the relationship between aerobic capacity measured separately in each leg and triathlon performance was not as high as in the respective single sports. A possible explanation for this finding may be that prior exercise affected the strength of the correlation between physiological variables specific to one discipline and performance in it under conditions characteristic of a triathlon competition [7,51].

### 4.2. Exercise Intensity during the Race

In the swimming leg, the mean relative intensities and the distribution of HR within the three intensity zones were similar between groups (Figure 2, Table 4). The mean relative HR of all participants was 89.8% of HR_max_, which is comparable to previous reports of elite triathletes (91%–92% of HR_max_) [22]. Surprisingly, the same level of relative HR (91.5% of HR_max_) was reported in the Ironman swimming leg (3.8 km, ~1 h) [52]. Although there is an obvious tendency toward reduced relative intensity with relation to increased race duration [20,28], this relationship may not necessarily be linear in triathlon swimming. Wu et al. [24] compared absolute exercise intensities across the three triathlon races (sprint, Olympic, and half-Ironman distance) performed by the same triathletes and found no differences in mean speed across the races despite differences in swimming duration (range, approximately 0:11:00–0:30:00 h:min:s). Therefore, relative intensity is considered comparable across the three triathlon races. Taken together, there might be little difference in relative intensities during the swimming leg at different performance levels and different race durations (~1 h). However, it is unclear whether about 90% of HR_max_ is the upper limit of sustainable intensity during triathlon swimming or if intensity is controlled based on anticipation of the longer duration of exercise following the swimming leg.

A visual inspection of the relative HR profile (Figure 1) suggests that the HR remained above HR at AnT throughout the swimming leg, as most of the leg was performed in HR_zone_3 (more than AnT), and the time spent in HR_zone_1 (less than AeT) was negligible (Figure 2), because HR_zone_1 was held only during the initial phase of the start of the race, when the triathletes began accelerating from the starting line. This is in agreement with a previous study demonstrating the HR distribution during 10-km cross-country skiing [19], during which the exercise duration (mean race time, 0:25:47 h:min:s) was comparable to the swimming leg in the present study (mean swimming time, 0:26:28 h:min:s).

Despite the stability of relative HR, large fluctuations above the AnT line were observed in the relative workload (Figure 1), which could be due to the influence of environmental factors. In open water swimming, the water conditions and tides can considerably affect absolute swimming speed and, therefore, relative swimming speed, even if the same relative HR is maintained.

In the cycling leg, all of the indicators of relative intensity were similar between groups (Figure 2 and Figure 3A, and Table 4). The mean relative HR of all participants was 91.1% of HR_max_. This value was nearly equal to those reported for elite triathletes during draft legal races (90%–92% of HR_max_) [21,22] and clearly higher than those reported for ironman triathletes (83% of peak HR) [26]. This value was slightly lower than the those reported for nonelite triathletes with comparable $\dot{V}$O_2 max_ as the subjects in the present study (60.6 vs. 58.7 mL·kg^−1^·min^−1^, respectively) during nondraft legal short-course races (94.1% HR_max_) [25], which may be partly due to some difference in the race distance of the cycling leg accompanied by differences in exercise duration between studies; i.e., the race distance and exercise duration were 30 km and 50.09 min in the previous study [25] and 40 km and 70.55 min in this study.

The mean relative workload in the cycling leg of all participants was 61.3% of PO_max_, which was nearly equal to that of elite triathletes during the draft legal race (60%–63% of PO_max_) [21,22]. However, some differences in the PO distribution evaluated with the same methodology were observed between the draft legal race in a previous study [22] and the nondraft legal race in the present study. In the former study [22], more time was spent in PO_zone_1 (below 10% of PO_max_) and PO_zone_5 (above PO_max_), with a comparatively shorter time in PO_zone_3 (between the PO at AeT and AnT) and PO_zone_4 (between the PO at AnT and PO_max_) in the present study. This discrepancy may be related to drafting, group riding dynamics, and the course profile. Elite triathletes often compete in criterium or circuit courses with frequent technical corners requiring repeated accelerations and decelerations with rotating the position within the same pack. Meanwhile, cycling during nondraft legal events is more similar to that of an individual time trial, as reflected by the stability of PO during flat cycling [53]. Therefore, the difference in the PO distribution may be related to the application rule of drafting and/or the course profile, even though the mean relative PO was similar in both races.

As shown by the relative HR profile in Figure 1, the HR remained above HR at AnT throughout the cycling leg. However, the PO profile (Figure 1) suggests that the PO shifted between PO at AeT and AnT throughout the leg. This discordance might be suggestive of the residual effect of prior swimming and the subsequent physiological responses that differ from performing an isolated exercise, especially in the initial phase [54]. Another explanation might be cardiovascular drift, which is primarily characterized by a progressive decrease in stroke volume and a progressive increase in HR to maintain cardiac output during prolonged exercise [55]. The hyperthermia and dehydration that occur concomitantly with long-duration exercise influence the magnitude of cardiovascular drift in a graded fashion [56]. Glycogen depletion caused by prolonged exercise also affects the workload–HR relationship [57]. Therefore, HR during an OD race may overestimate the PO [58]. Even in the running leg, the relative HR level was higher than the relative workload level.

In the running leg, there were clear differences in relative intensity between the faster and slower groups. The mean relative HR was definitely higher in the faster group than the slower group (94.0% vs. 87.0% of HR_max_) and slightly higher than the recreational triathletes (92.3% HR_max_) [25], but on a comparable level for elite triathletes (93%–94% of HR_max_) [22]. Additionally, the faster group spent more time in the high-intensity zones (i.e., larger values of HR_zone_3 and RS_zone_4) as compared to the slower group. The relative workload was also higher in the faster group than the slower group (83.4% vs. 77.2% of RS_max_). Therefore, our hypothesis is supported only in the running leg.

It is interesting to note that clear differences in relative intensity between the two groups were observed only in the running leg in this study. In terms of the pacing strategy, it has been demonstrated that the slower, less experienced athletes tended to pace at too high an intensity at the beginning of the race, leading to a continuous decrease in exercise intensity for the remaining duration [32]. However, in the swimming and cycling legs in this study, there were no differences in any indicator of relative intensity or triathlon experience between the two groups. Thus, the effect of the height and distribution of relative intensity until the completion of the cycling leg may have had only a minor impact on the difference in relative intensity during the running leg between the two groups. On the other hand, before the start of the running leg, the slower group was exposed a longer duration of relative intensity (~10 min) at a level comparable to that of the faster group. This difference might have induced greater physiological and metabolic disruptions in the slower group, resulting in premature fatigue and impaired workload in the running leg.

Millet and Vleck [7] stated that the first transition (i.e., swim–cycle transition) is regarded as having a negligible effect on overall performance during the nondraft legal OD race. Meanwhile, a number of studies reported the effects of prior exercise on subsequent running [7,8,59,60,61] and the cycle–run transition is traditionally considered to be more important on performance during an OD race [7,8]. Prior cycling induces physiological/cardiorespiratory and biomechanical changes during subsequent running [8]. Several studies have reported that such changes (usually negative effects) might be related to the performance level of triathletes [62,63,64,65,66]. For example, Boussana et al. [62] observed significantly higher ventilatory responses and significantly greater decreases in respiratory muscle strength and endurance in competition triathletes than in the elite group in running following cycling performed at similar relative intensities, despite similar $\dot{V}$O_2 max_ and VT [62]. Therefore, in this study, the negative effects of prior exercise on physiological and biomechanical responses during subsequent running might have been larger in the slower group than the faster group, which could be related to the difference in relative intensity between the groups during the running leg. Further studies are required to specifically analyze the mechanisms underlying the difference in relative intensity induced by prior exercise.

From the point of view of exercise intensity throughout the OD race, relative HR remained above HR at AnT until the end of the cycling leg and remained at or above HR at AnT in the running leg in both groups. These values were higher than those previously reported for ultra-endurance events (race durations >8 h) [26,27] and a 42-km marathon (~2.5–5 h), during which intensity was approximated as AeT [30]. A possible explanation for the higher intensity during the OD race as compared to marathon―both with similar estimated energy expenditures (2000–2546 kcal) [67]―may lie in the degree of muscle fatigue derived from peripheral factors. Although the causes of muscle fatigue are complex and not completely understood, exercise-induced muscle damage is a primary cause of muscle fatigue during endurance sports [68]. While no direct comparison is available, expression levels of markers of muscle damage, such as creatine kinase and lactate dehydrogenase, are higher after marathon running [68] than after an OD triathlon [10]. Additionally, a reduction in the height of countermovement jump, which is also higher after a marathon [10,68], suggesting that exercise-induced fatigue of lower extremity muscles is greater while competing in a marathon as compared to an OD triathlon. Swimming and cycling are considered to produce minor damage to the involved muscles, while running―a weight-bearing exercise that includes concentric and eccentric contractions of the leg muscles―may produce more pronounced muscle damage while competing in a 42-km marathon. Another possible explanation could be related to the degree of glycogen depletion, as proposed in a previous report [69]. Glycogen depletion is also related to muscle fatigue during prolonged exercise [70]. Differences in muscle activities among the three exercise modes might be related to higher glycogen reserves in active muscles in each discipline of a triathlon as compared to strictly running in a marathon. However, further investigations are needed to elucidate the specific physiological responses during a triathlon.

### 4.3. Limitations

There were several methodological limitations to this study that should be acknowledged. First, the competition measurements were carried out in several separate OD races. This disparity may have added some bias to the analysis of the present study. However, there were no significant differences in environmental conditions or course profiles between the two groups (Table 2), and thus, the effects of these disparities were considered minor. Moreover, differences in the swimming conditions between the laboratory tests (i.e., flume swimming) and competition measurements (i.e., open water swimming) could affect the analysis of this study. Also, the air temperature in the laboratory tests (around 25 °C) was higher than that in the OD races (around 21.6 °C). Cardiovascular drift can occur in temperate conditions and greater effects are seen in high heat conditions [71], so the difference mentioned above could impact the $\dot{V}$O_2_ or BLa vs. HR relationships. However, an electric fan provided airflow, and overall room ventilation was maintained throughout each test, which minimizes the possibility of a cardiovascular drift. Furthermore, each incremental test was performed at 45 ± 26 days from the competition measurements of the OD race in this study, while most previous similar studies performed incremental tests within 2 to 4 weeks before the OD race [21,22,25,72]. The individual training programs and different season calendars among our subjects made it difficult to arrive at a unified schedule in this study. Although the maximal and submaximal measures of triathletes are reportedly stable throughout the pre-competition to competitive period [73], future research examining exercise intensity during OD triathlon should attempt to use unified race conditions and schedules.

## 5. Conclusions

Mean exercise intensity during a nondraft legal OD triathlon was above 87% of HR_max_ derived from each exercise mode throughout all three legs in well-trained, age-group male triathletes. The majority of the swimming and cycling legs was spent at an intensity more than HR at AnT, while the exercise intensity during the running leg differed among individuals. The time to complete the whole OD race showed that the intensity of the faster triathletes was higher than that of the slower triathletes. The results of the present study suggest that sustaining higher intensity during the running leg might be important for success in nondraft legal OD triathlon races. These results may be beneficial for athletes, coaches, and researchers in the sense that they describe the characteristic performance profiles of the multisport nature of triathlon events. As such, the present research may give rise to better plan training and racing strategies.

## Figures and Tables

**Figure 1 sports-09-00018-f001:**
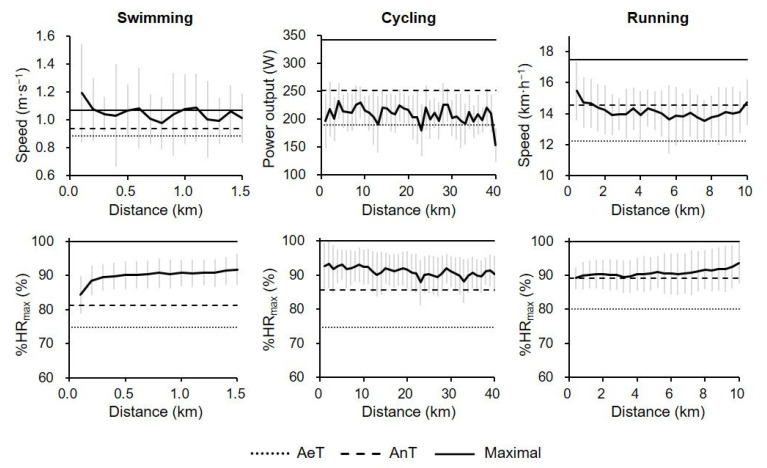
Profile of the percentage of maximal heart rate (%HR_max_) and absolute workload (swimming speed, cycling power output, and running speed) in each leg during Olympic-distance races (*N* = 17). **Upper figures**: mean %HR_max_ at AeT (dotted line), %HR_max_ at AnT (dashed line), and HR_max_ (solid line). **Lower figures**: mean workload at AeT (dotted line), workload at AnT (dashed line), and maximal workload (solid line). See text for explanations of AeT, AnT, and maximal workload in each exercise mode.

**Figure 2 sports-09-00018-f002:**
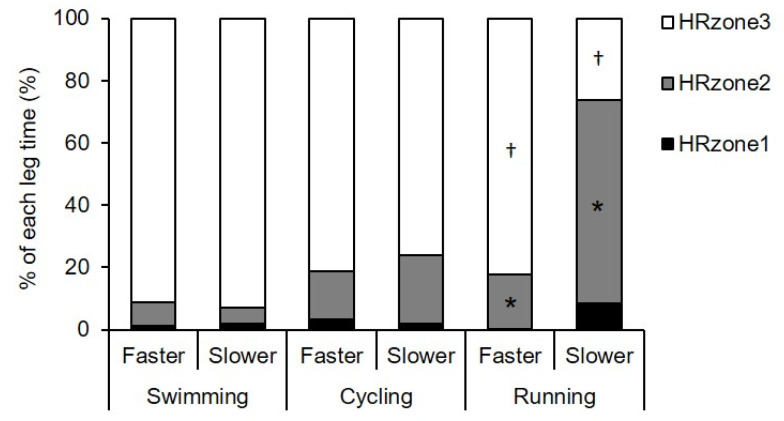
Time percentage in three intensity zones based on heart rate (HR) in each leg during Olympic-distance races in the faster (*n* = 9) and slower (*n* = 8) groups. * *p* < 0.05 in HR_zone_2 in running leg. † *p* < 0.05 in HR_zone_3 in running leg. Standard deviations have been eliminated to improve clarity. See text for explanations of HR_zone_1, 2, and 3.

**Figure 3 sports-09-00018-f003:**
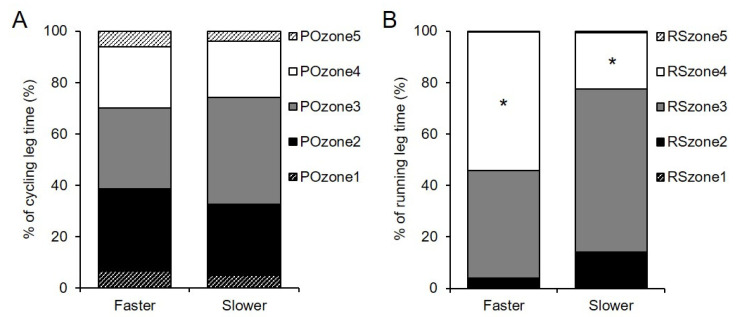
Time percentage in five intensity zones based on power output (PO, **A**) in the cycling leg and running speed (RS, **B**) in the running leg during Olympic-distance races in the faster (*n* = 9) and slower (*n* = 8) groups. No significant differences were observed between the groups in each PO_zone_ in the cycling leg (*p* > 0.05). * *p* < 0.05 in RS_zone_4 in the running leg. Standard deviations have been eliminated to improve clarity. See text for explanations of each PO_zone_ and RS_zone_.

**Table 1 sports-09-00018-t001:** Characteristics and Olympic-distance race times of the faster and slower groups.

	All (*N* = 17)	Faster (*n* = 9)	Slower (*n* = 8)
Subject characteristics			
Age (yr)	23.1 ± 6.7	24.3 ± 8.4	21.6 ± 4.2
Height (cm)	173.8 ± 5.9	174.2 ± 5.2	173.4 ± 7.0
Mass (kg)	65.1 ± 5.5	64.9 ± 6.1	65.3 ± 5.2
Body Fat (%)	10.3 ± 1.7	9.8 ± 1.4	10.9 ± 1.9
BMI	21.5 ± 1.1	21.4 ± 1.5	21.7 ± 0.3
Triathlon experience (yr)	4.1 ± 6.1	5.4 ± 8.1	2.6 ± 2.4
Olympic-distance race times			
Swimming (h:min:s)	0:26:28 ± 0:04:01	0:23:34 ± 0:01:49	0:29:45 ± 0:03:10 **
Cycling (h:min:s)	1:10:33 ± 0:02:53	1:08:54 ± 0:02:19	1:12:25 ± 0:02:19 *
Running (h:min:s)	0:42:49 ± 0:04:39	0:39:57 ± 0:02:14	0:46:02 ± 0:04:37 **
Total (h:min:s)	2:19:50 ± 0:09:38	2:12:24 ± 0:02:54	2:28:12 ± 0:07:11 **

Values are means ± SD. *N*, number of subjects; BMI, body mass index. The nonparametric Mann–Whitney U test was used to detect statistically significant differences between the groups. * *p* < 0.05; ** *p* < 0.01.

**Table 2 sports-09-00018-t002:** Environmental conditions during the Olympic-distance race of the faster and slower groups.

	All (*N* = 17)	Faster (*n* = 9)	Slower (*n* = 8)
Overall			
Air temperature (°C)	21.6 ± 3.7	20.9 ± 3.6	22.3 ± 3.9
Relative humidity (%)	64.3 ± 17.0	58.6 ± 15.2	70.0 ± 17.7
Wind speed (m·s^−1^)	3.5 ± 1.0	3.4 ± 0.9	3.6 ± 1.1
Barometric pressure (mmHg)	759.1 ± 7.1	760.8 ± 6.1	757.4 ± 8.1
Swimming			
Water temperature (°C)	20.3 ± 3.3	21.3 ± 3.1	20.1 ± 3.7
Cycling			
Cumulated positive elevation (m)	173.7 ± 115.6	147.8 ± 134.2	202.8 ± 90.3
Elevation to distance ratio (m·km^−1^)	4.3 ± 2.9	3.7 ± 3.4	5.1 ± 2.3
Running			
Cumulated positive elevation (m)	37.7 ± 31.3	28.7 ± 20.6	47.8 ± 39.2
Elevation to distance ratio (m·km^−1^)	3.8 ± 3.1	2.9 ± 2.1	4.8 ± 3.9

Values are means ± SD. *N*, number of subjects. No significant differences were observed between the groups (*p* < 0.05).

**Table 3 sports-09-00018-t003:** Laboratory measurements of swimming, cycle ergometry, and treadmill running of the faster and slower groups.

	All (*N* = 17)	Faster (*n* = 9)	Slower (*n* = 8)
Swimming			
Speed at AeT (m·s^−1^)	0.88 ± 0.14	0.98 ± 0.10	0.78 ± 0.10 **
Speed at AnT (m·s^−1^)	0.93 ± 0.13	1.02 ± 0.10	0.84 ± 0.09 **
SS_max_ (m·s^−1^)	1.07 ± 0.13	1.16 ± 0.09	0.96 ± 0.08 **
HR at AeT (bpm)	138 ± 17	142 ± 15	135 ± 19
HR at AnT (bpm)	150 ± 13	149 ± 10	151 ± 18
HR_max_ (bpm)	185 ± 9	186 ± 9	182 ± 10
%HR_max_ at AeT (%)	74.8 ± 6.7	76.0 ± 6.8	73.6 ± 6.7
%HR_max_ at AnT (%)	81.2 ± 6.0	80.0 ± 4.8	82.5 ± 7.2
Cycling			
PO at AeT (W)	190 ± 32	203 ± 28	176 ± 33 *
PO at AnT (W)	252 ± 33	263 ± 32	239 ± 31
PO_max_ (W)	343 ± 34	359 ± 34	325 ± 24 *
HR at AeT (bpm)	136 ± 15	140 ± 16	132 ± 14
HR at AnT (bpm)	156 ± 13	158 ± 16	154 ± 10
HR_max_ (bpm)	183 ± 8	183 ± 9	182 ± 7
%HR_max_ at AeT (%)	74.7 ± 6.5	76.6 ± 6.3	72.6 ± 6.5
%HR_max_ at AnT (%)	85.6 ± 5.0	86.5 ± 6.2	84.6 ± 3.4
$\dot{V}$O_2 max_ (L·min^−1^)	3.8 ± 0.4	4.0 ± 0.3	3.6 ± 0.3 *
$\dot{V}$O_2 max_ (ml·kg^−1^·min^−1^)	58.7 ± 5.9	62.4 ± 4.7	54.5 ± 3.9 **
% $\dot{V}$O_2 max_ at AeT (%)	63.7 ± 7.9	64.7 ± 5.5	62.6 ± 10.2
% $\dot{V}$O_2 max_ at AnT (%)	80.8 ± 6.2	81.2 ± 7.5	80.3 ± 4.8
GE (%)	21.3 ± 1.4	21.3 ± 1.2	21.3 ± 1.6
Running			
Speed at AeT (km·h^−1^)	12.2 ± 1.2	12.6 ± 1.0	11.9 ± 1.3
Speed at AnT (km·h^−1^)	14.5 ± 1.5	14.9 ± 0.9	14.1 ± 2.0
RS_max_ (km·h^−1^)	17.5 ± 1.0	17.8 ± 0.7	17.1 ± 1.2
HR at AeT (bpm)	154 ± 10	153 ± 13	155 ± 8
HR at AnT (bpm)	171 ± 13	171 ± 13	172 ± 12
HR_max_ (bpm)	192 ± 9	190 ± 9	194 ± 9
%HR_max_ at AeT (%)	80.1 ± 4.0	80.3 ± 4.6	79.8 ± 3.5
%HR_max_ at AnT (%)	89.1 ± 4.4	89.6 ± 3.9	88.5 ± 5.1
$\dot{V}$O_2 max_ (L·min^−1^)	3.9 ± 0.4	4.0 ± 0.4	3.8 ± 0.4
$\dot{V}$O_2 max_ (ml·kg^−1^·min^−1^)	60.4 ± 4.4	62.2 ± 2.8	58.5 ± 5.3
% $\dot{V}$O_2 max_ at AeT (%)	72.2 ± 6.1	71.8 ± 5.5	72.8 ± 7.0
% $\dot{V}$O_2 max_ at AnT (%)	86.9 ± 6.2	86.2 ± 3.9	87.6 ± 8.3
Running economy (ml·kg^−1^·km^−1^)	217 ± 14	217 ± 9	216 ± 18

Values are means ± SD. *N*, number of subjects; AeT, aerobic threshold; AnT, anaerobic threshold; SS_max_, Maximal swimming speed; HR, heart rate; HR_max_, maximal heart rate; PO, power output; PO_max_, Maximal power output, $\dot{V}$O_2 max_, maximal oxygen uptake; GE, gross efficiency; RS_max_, Maximal running speed. * *p* < 0.05; ** *p* < 0.01. See text for explanations of AnT and AeT in each exercise mode.

**Table 4 sports-09-00018-t004:** Mean absolute HR, mean relative HR, mean absolute workload, and mean relative workload of the faster (*n* = 9) and slower (*n* = 8) groups.

	Group	Swimming	Cycling	Running
Absolute workload ^a^	All (*N* = 17)	1.03 ± 0.18	210 ± 24	14.1 ± 1.4
	Faster (*n* = 9)	1.14 ± 0.18	221 ± 26	14.9 ± 0.8
	Slower (*n* = 8)	0.92 ± 0.09 **	197 ± 15	13.2 ± 1.4 *
Relative workload (%maximal workload)	All (*N* = 17)	96.6 ± 8.8	61.3 ± 5.2	80.5 ± 4.9
	Faster (*n* = 9)	97.8 ± 10.0	61.5 ± 5.5	83.4 ± 2.9
	Slower (*n* = 8)	95.2 ± 7.7	61.0 ± 5.2	77.2 ± 4.5 *
Absolute HR (bpm)	All (*N* = 17)	166 ± 12	166 ± 9	174 ± 9
	Faster (*n* = 9)	170 ± 13	170 ± 7	179 ± 7
	Slower (*n* = 8)	162 ± 11	162 ± 8	169 ± 9 *
Relative HR (%HR_max_)	All (*N* = 17)	89.8 ± 3.7	91.1 ± 4.4	90.7 ± 5.1
	Faster (*n* = 9)	90.9 ± 3.1	93.1 ± 4.3	94.0 ± 2.2
	Slower (*n* = 8)	88.6 ± 4.1	89.0 ± 3.7	87.0 ± 5.1 **

Values are means ± SD. *N*, number of subjects; HR, heart rate; HR_max_, maximal heart rate. * *p* < 0.05; ** *p* < 0.01. ^a^ Swimming, speed (m·s^−1^); Cycling, power output (W); Running, speed (km·h^−1^).

## Data Availability

The data presented in this study are available on request from the corresponding author.

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
