# Peer review of "Exercise Intensity during Olympic-Distance Triathlon in Well-Trained Age-Group Athletes: An Observational Study"

_sports, 2021, doi:10.3390/sports9020018_

Round 1
Reviewer 1 Report
Accurately profiling the physiological capacity is 3x the challenge compared to single discipline performers. My overall impression of this study, both in methodological execution and in the written presentation was very positive. I was very impressed with the quality of English, and found myself wondering if there was a native English speaker in the author list that I was not seeing until near the end of the discussion when I actually detected some small grammatical errors!
This is a data-rich study and provides the reader with a lot of "typical response data" that I find useful. I actually have few methodological/interpretation issues to pick at. However, there are a couple I will mention:
1. The laboratory tests appear to have been performed at 25C. This is a temperature where we start to see an accelerated cardiac drift, particularly if airflow across the skin and overall room ventilation is not excellent. The lab conditions are warmer than the conditions during racing, and this could impact the HR/workload relationships that you are attempting to establish using lab tests as calibrations. The authors should briefly comment on this issue.
2. We generally see significant cardiac decoupling across time during long endurance events. At the same time, the scope of decoupling that can be observed is reduced when the work intensity is at or above the second lactate turn point. You do mention cardiac drift in lines 447-456, but I think it is important to emphasize that HR as a surrogate for actual work intensity will tend to overestimate external workload as fatigue and glycogen depletion develop. The relative HR drift is also difficult to compare across individuals without correcting for differences in actual heart rate reserve (HRmax minus HR rest)
3. I think it would be relevant to address differences in the swimming conditions between swimming in a flume and swimming in a pack or "line" during the triathlon. You take up this issue for cycling, but the HR-swimming speed relationship could also be impacted both negatively (chaotic, splashing, etc) and positively (drafting effect in the water).
Indeed, for beginning/recreational triathletes, my understanding is that the swim is the most dangerous part of the race is the swim and numerous cardiac events have been reported during this stage when competitors are subjected to "near-drowning" experiences.
Reviewer 2 Report
Sports-1064196
Reviewer’s Comments
In the submitted manuscript, the authors study the exercise intensity parameters in the three separate events that comprise the Olympic triathlon. Seventeen male triathletes completed incremental swimming, cycling, and running tests to exhaustion in the laboratory to examine various physiological parameters such as the heart rate and the workloads at an intensity corresponding to the aerobic and anaerobic thresholds. Results revealed that the mean percentage of HRmax and intensity distributions during the swimming and cycling legs were similar between the faster and slower participants. In addition, the faster group spent relatively more time above HR at anaerobic threshold and between workload at anaerobic threshold and maximal workload. It is concluded that well-trained male triathletes performed at very high intensity throughout the triathlon race and that keeping a higher intensity during running might be essential concerning success.
The research is within the scope of the Journal. It is well written and it can be considered for publication after addressing the topics that are mentioned in the General and Specific Comments.
General comments
- Materials & Methods:
- The participants were dichotomized into the two groups? What was the difference in performance between the slowest in the fast group and the faster in the slow group?
- Incremental tests: provide the rationale of selecting an incremental vs. other pace related tests? It is suggested to provide the rationale and the relative information about pacing in the Introduction.
- In addition, see Poole et al. (2020) [ Poole, D. C., Rossiter, H. B., Brooks, G. A., & Gladden, L. B. (2020). The anaerobic threshold: 50+ years of controversy. Journal of Physiology, doi: 10.1113/JP280980] for the definitions used concerning the exercise intensity parameters.
- Statistical analysis: is it meaningful to examine a possible group X leg effect and/or interaction?
- It is suggested to type the abbreviations as one word (for example: HRzone2 instead of HRzone 2).
- It is also suggested to use an uniform manner to state time (i.e. h:min:s and not as 26 min 28 s as seen in subsection 4.2).
Specific comments
Abstract
- L13: What do you mean “age-group”? The same in L24, L101 and elsewhere.
Introduction
- See the respective comment about the incremental tests and pacing.
- L41-43: Elaborate on why is this necessary to discern.
Materials and Methods
- See the respective General Comment about the incremental tests, the participants and the statistical analysis.
- L109: For how long prior the experiment was the ‘injury free’ state considered as inclusive criteria?
- L116: State the statistical test used for the comparison of the two groups presented in Table 1.
- L118-125: It is suggested to move this subsection prior subsection 2.1 Subjects
- L246-250: It is suggested to integrate this subsection in subsection 2.1 Subjects
Results
- See the respective General Comment about the statistical analysis and the abbreviations.
Discussion
- See the respective General Comment about the abbreviations and the presentation of time.
- L402: range: 11-30 min.
- L524: Use other term rather than “noise”.
Conclusions
- Add a specific training suggestion based on the findings of the present study.
References
- Use the abbreviated Journal title for references 3, 4, 14, 29, 30, 51, 65, 70.
